# β-Nicotinamide Mononucleotide Alleviates Hydrogen Peroxide-Induced Cell Cycle Arrest and Death in Ovarian Granulosa Cells

**DOI:** 10.3390/ijms242115666

**Published:** 2023-10-27

**Authors:** Yunduan Wang, Qiao Li, Zifeng Ma, Hongmei Xu, Feiyu Peng, Bin Chen, Bo Ma, Linmei Qin, Jiachen Lan, Yueyue Li, Daoliang Lan, Jian Li, Shujin Wang, Wei Fu

**Affiliations:** 1College of Animal & Veterinary Sciences, Southwest Minzu University, Chendu 610041, China; wangyunduan@stu.swun.edu.cn (Y.W.); hongmeixu99@163.com (H.X.); l494382646@163.com (J.L.);; 2Key Laboratory of Qinghai-Tibetan Plateau Animal Genetic Resource Reservation and Utilization, Southwest Minzu University, Ministry of Education, Chengdu 610041, China; 3Institute of Life Sciences, Chongqing Medical University, Chongqing 400032, China; 4Key Laboratory of Animal Science of National Ethnic Affairs Commission of China, Southwest Minzu University, Chengdu 610041, China

**Keywords:** hydrogen peroxide, ovarian granulosa cells, β-nicotinamide mononucleotide, cell cycle, cell death

## Abstract

Maintaining normal functions of ovarian granulosa cells (GCs) is essential for oocyte development and maturation. The dysfunction of GCs impairs nutrition supply and estrogen secretion by follicles, thus negatively affecting the breeding capacity of farm animals. Impaired GCs is generally associated with declines in Nicotinamide adenine dinucleotide (NAD^+^) levels, which triggers un-controlled oxidative stress, and the oxidative stress, thus, attack the subcellular structures and cause cell damage. β-nicotinamide mononucleotide (NMN), a NAD^+^ precursor, has demonstrated well-known antioxidant properties in several studies. In this study, using two types of ovarian GCs (mouse GCs (mGCs) and human granulosa cell line (KGN)) as cell models, we aimed to investigate the potential effects of NMN on gene expression patterns and antioxidant capacity of both mGCs and KGN that were exposed to hydrogen peroxide (H_2_O_2_). As shown in results of the study, mGCs that were exposed to H_2_O_2_ significantly altered the gene expression patterns, with 428 differentially expressed genes (DEGs) when compared with those of the control group. Furthermore, adding NMN to H_2_O_2_-cultured mGCs displayed 621 DEGs. The functional enrichment analysis revealed that DEGs were mainly enriched in key pathways like cell cycle, senescence, and cell death. Using RT-qPCR, CCK8, and β-galactosidase staining, we found that H_2_O_2_ exposure on mGCs obviously reduced cell activity/mRNA expressions of antioxidant genes, inhibited cell proliferation, and induced cellular senescence. Notably, NMN supplementation partially prevented these H_2_O_2_-induced abnormalities. Moreover, these similar beneficial effects of NMN on antioxidant capacity were confirmed in the KGN cell models that were exposed to H_2_O_2_. Taken together, the present results demonstrate that NMN supplementation protects against H_2_O_2_-induced impairments in gene expression pattern, cell cycle arrest, and cell death in ovarian GCs through boosting NAD^+^ levels and provide potential strategies to ameliorate uncontrolled oxidative stress in ovarian GCs.

## 1. Introduction

Granulosa cells (GCs) are of the most importance for maintaining follicular development in the ovary as they provide nutrients and gonadal hormones to support oocyte development [1]. After the initiation of primordial follicle development, a few flattened somatic cells surrounding the oocytes multiply to form several layers and the cell numbers increase [2,3]. In preantral follicles, those somatic cells have the potential to differentiate into two types of cells: cumulus cells (CCs) and parietal GCs [2,3,4]. The GCs contribute to follicular development or atresia by regulating gonadotropin receptors, steroid hormones, various growth factors, and cytokines [5,6]. Therefore, the natural proliferation and differentiation of GCs determine the initiation, maturation, ovulation, steroid hormone synthesis, and secretion of follicular development [7].

Matured GCs directly regulate the development of oocytes, and then maintain the function of the ovary. However, an abnormal apoptosis of GCs will trigger atresia [8]. Growing evidence from in vitro and in vivo studies have shown that GCs are very susceptible to oxidative stress, which leads to cellular oxidative damage [9,10]. In follicular microenvironment, GCs communicate with oocytes via gap junctions to supply nutrients for oocytes development and maturation. This connection can be blocked when GCs suffer from uncontrolled oxidative stress, and then leads to developmental arrest of follicles [11]. Oxidative stress is attributed to the imbalance of the intracellular oxidative and antioxidant system. The production rate of reactive oxygen species (ROS) is greater than their scavenging rate, which results in massive accumulation of ROS in cells, eventually attacking the subcellular structures to cause cellular damage [12]. In addition to follicular atresia, oxidative stress can cause a variety of ovarian-related diseases (e.g., endocrine disorders and premature ovarian failure), which seriously endanger the reproductive health of farm animals and human beings [13].

Hydrogen peroxide (H_2_O_2_), a potent oxidant that is generally considered as causing in vitro induction of oxidative stress, is employed to investigate the underlying mechanism of uncontrolled oxidative stress on GCs functions. Several studies have shown that H_2_O_2_ exposure induces apoptosis [14,15] and autophagy [16,17] and inhibits the steroid synthesis function of GCs [18]. Accordingly, a variety of antioxidants, such as melatonin [9], resveratrol [10], vitamins, and selenium [19], can effectively alleviate H_2_O_2_-induced oxidative stress in GCs. Furthermore, some key signaling pathways, including p53 [14], JNK, mTOR, PI3K [16], SIRT [10], NRF, and MAPK [20], have been proven to be involved in the regulation of these processes. Notably, H_2_O_2_-triggered oxidative stress and cell death is mainly due to decreased levels of nicotinamide adenine dinucleotide (NAD^+^), which is a key signaling molecule that controls cellular function in environmental changes, like nutrient intake and cellular damage [21]. The levels of NAD^+^ in the body gradually decreases with aging, thus leading to the functional decline of cells and organs, metabolic abnormalities, and the increased susceptibility to diseases [11]. Accordingly, we are in search for potential strategies for boosting NAD^+^ levels to protect against ovarian-related diseases through inhibiting oxidative stress. The nicotinamide mononucleotide (NMN), as a NAD^+^ precursor, is confirmed to be a non-toxic additive, which works as the most direct and effective NAD^+^ supplement [12]. Interestingly, NMN can partially alleviate the aging-induced depletion of female reproductive capacity [22]. Although some studies have demonstrated that NMN supplementation can improve H_2_O_2_-induced abnormalities in other cell models [23], there is no report on the potential effects of NMN on H_2_O_2_-induced oxidative stress in GCs. In this study, the two in vitro GCs models (mouse ovarian granulosa cells (mGCs) and human granulosa cell lines (KGN)) were employed to investigate the effect of NMN supplementation on gene expression patterns, antioxidant capacity, and cell death of both mGCs and KGN that were exposed to H_2_O_2_.

## 2. Results

### 2.1. NMN Supplementation Shapes the Gene Expression Pattern in mGCs That Are Exposed to H_2_O_2_

Firstly, we conducted a principal components analysis (PCA) among group C (control), group H (150 μM H_2_O_2_), and group HN (150 μM H_2_O_2_ + 500 μM NMN). According to the analysis, the first principal component (PC1) and the second principal component (PC2) accounted for 83.3% and 10.5%, respectively. Moreover, the contribution of PCA to the sample difference was more than 93%, which could represent clear differences among different conditions (Figure 1A). Meanwhile, RNA-seq analysis results showed that the distance was relatively close between group H and HN, which indicate that NMN could partly recover the gene expression in H_2_O_2_-cultured mGCs. The cluster analysis also showed that group C significantly differed from group H and group HN and suggested that the gene expression pattern in mGCs were affected by H_2_O_2_ or NMN (Figure 1B).

The gene distribution with expression abundance (FPKM) > 1 in the three groups was plotted by a Venn diagram (Figure 1C). The threshold of differential genes was set to |Log_2_ (FC)| > 1 and Q value < 0.05 (the same below). Furthermore, the differentially expressed genes (DEGs) between the groups were counted in Figure 1D. Accordingly, 155 up-regulated and 273 down-regulated genes were detected in group C vs. group H, respectively. Furthermore, the number of up-regulated and down-regulated genes in group C vs. group HN significantly increased to 231 and 390, respectively (Appendix A).

Next, specific DEGs were used for verifying the reliability of RNA-seq data by the RT-qPCR method. Four differential genes, including *Trib3*, *Ccng1*, *Ddit3*, and *Col11a1*, were randomly screened for this verification. The results of RT-qPCR showed that the gene expression trend was consistent with the results of RNA-seq analysis (Figure 1E), thereby indicating the high reliability of RNA-seq data in this study.

### 2.2. The Transcriptome Analysis of mGCs That Are Exposed to H_2_O_2_

The cluster analysis was used for detecting the DEGs between group C and group H, and we found that the two groups formed two independent clusters (Figure 2A). According to the threshold set above, the number of DEGs was 428, of which 155 were up-regulated and 273 were down-regulated in group H. The DEGs are shown in the heatmap (Figure 2B). Furthermore, Gene Ontology (GO), Kyoto Encyclopedia of Genes and Genomes (KEGG), Reactome database analysis, and Gene Set Enrichment Analysis (GSEA) were then performed to analyze the DEGs (Figure 2 and Figure 3).

Firstly, the DEGs were classified into three groups, including biological process (BP), molecular function (MF), and cellular component (CC) by GO terms using the online database and further clustered based on gene functions and signaling pathways, respectively. Among them, the top three terms of BP were cellular process, single-organism process, and metabolic process. Moreover, the top three terms of MF were binding, catalytic activitym and transporter activity, while cell part, cell, and organelle belonged to the top three terms of CC (Figure 2C). Furthermore, the top ten items in GO were mainly related to cell cycle process and mitotic cycle process (Figure 2D), and this indicate that H_2_O_2_-induced changes in gene expression patterns were highly involved in the cell cycle (Appendix A).

Secondly, the KEGG pathway analysis showed that DEGs between group C and group H were mainly classified into the following major items: metabolism, human diseases, organismal systems, genetic information processing, cellular processes, and environmental information processing (Figure 3A). Notably, the KEGG pathway enrichment analysis further revealed that those DEGs were mainly enriched in the cell cycle and P53 pathway (Figure 3B). With respect to this, growing evidence suggested that P53 was highly linked with regulating cell proliferation and division [24], and these findings of KEGG analysis were consistent with the GO analysis above. More importantly, those DEGs were also enriched in oocyte meiosis and progesterone-mediated oocyte maturation (Figure 3B), and this implies that H_2_O_2_-induced oxidative stress in mGCs not only affected the cell cycle process, but also contributed to the development and maturation of oocytes (Appendix A).

Moreover, the Reactome database was used for analyzing those DEGs, and we observed that the main enriched items were still associated with cell cycle process, mitotic, and cell cycle checkpoints (Figure 3C). Besides, these pathways were enriched in centrosomes and kinetochore, which were essential for cell division (Appendix A). Furthermore, by using GESA analysis, we analyzed the expression of all genes in mGCs exposed to H_2_O_2_, and then mapped the obtained top ten items. According to these results, these main items enriched by GO-GSEA were chromosome segregation, mitotic cell cycle process, and cell cycle process (Figure 3D), whereas those enriched by KEGG-GSEA were related to DNA replication and cell cycle (Figure 3E). Interestingly, all enriched items are also closely related to the cell cycle.

The STRING protein interaction database was also employed to analyze the target protein interaction network of DEGs in H_2_O_2_-cultured mGC. Accordingly, the protein interaction analysis of DEGs that were enriched in the KEGG pathways, such as cell cycle progression, DNA replication, and P53 signaling, showed that *Cdk1* and *Cdkn1a* genes were located at the center of the interaction network map, thus suggesting that the two genes were the core pathway of the above-mentioned pathways (Figure 3F,G). The two hub genes, *Cdk1* and *Cdkn1a*, were highly correlated with cell division cycle-related genes (e.g., *Cdc27*), cyclin family (e.g., *Ccnd2* and *Ccnd3*), proliferating cell nuclear antigen (*PCNA*), and *TP53*. These findings further approved that H_2_O_2_-triggered uncontrolled oxidative stress and the cell cycle arrest by H_2_O_2_ exposure was closely linked with significant changes in the gene expression pattern of mGCs.

### 2.3. H_2_O_2_ Induces Ovarian Granulosa Cell Cycle Arrest and Apoptosis

On the basis of the results of the RNA-seq analysis described above, the CCK-8 kit was used for determining the cell activity of mGCs that were treated with different concentrations of H_2_O_2_ (0, 50, 150, 300 μM). The cell viability rate of mGC presented a significant decrease in a H_2_O_2_ concentration-dependent manner (Figure 4A). Next, the cell status of H_2_O_2_-cultured mGCs was analyzed, and we found that when compared with the control group, the cells in group H had a larger volume as well as lower numbers (Figure 4B). Furthermore, using SA-β-Gal staining, we found that H_2_O_2_-mGCs had higher numbers of β-galactosidase, and this indicate that H_2_O_2_ caused the senescence and apoptosis in cells (Figure 4C).

Next, the results of RT-qPCR also confirmed that *Gsta4* (of the antioxidant gene glutathione family), *Bax* (an apoptosis marker) (Figure 4D), *Cdkn1a* (an apoptosis marker), and *TrpP53input* and *Hmox1* (key oxidative stress resistance molecules) were significantly up-regulated in group H when compared with those of group C (Figure 4E), whereas the cell cycle and proliferation related factors, including *Ki67*, *Pcna*, *Ccnd1*, and *Cdk1*, were distinctly down-regulated with H_2_O_2_ exposure (Figure 4F).

In addition, human ovarian granulosa cells (KGN) were cultured with different concentrations of H_2_O_2_ (100, 200, 300, 400, 500 μM). The results of bright field microscopy showed that the survival rate of KGN cells gradually decreased with increased concentrations of H_2_O_2_ (Figure 4G). Specifically, the number of adherent living cells gradually decreased with increasing H_2_O_2_ concentrations. Furthermore, the cells in all groups were stained with a PI staining kit, and the number of living cells was also counted (Figure 4H,I). Accordingly, the number of KGN living cells significantly decreased with increased concentrations of H_2_O_2_. Notably, the statistical results from cells that were exposed to 200 μM H_2_O_2_ showed that the number of living cells was significantly reduced to approximately 1/3 of the control group.

Generally, the addition of H_2_O_2_ contributed to disturbances of the gene expression pattern in ovarian GCs. Of note, these DEGs were mainly involved in the cell cycle, apoptosis, oxidative stress damage, and development processes.

### 2.4. Transcriptome Analysis of mGCs Induced by the Co-Addition of H_2_O_2_ and NMN

The results above indicate that H_2_O_2_ negatively affected the cell cycle progression by changing the gene expression pattern in mGCs. Given that NMN protects against oxidative stress and apoptosis of cells [22,23], we then investigated the effects of NMN supplementation on gene expression patterns and cell cycle in H_2_O_2_-cultured mGCs. Accordingly, when compared with group C (control group), group HN (150 μM H_2_O_2_ + 500 μM NMN) was clustered separately, and this indicate there was a significant difference between the two groups (Figure 5A). Furthermore, 621 DEGs were observed between the two groups. Among them, 231 up-regulated DEGs and 390 down-regulated DEGs in group HN were significantly higher than those of group C and group H (Figure 5B).

Next, the GO enrichment of these DEGs in mGCs was also found to be mainly enriched in the cell cycle process, cell cycle, and mitotic cycle process (Figure 5C; Appendix A). Being consistent with the above-mentioned findings, all these enriched items were involved in the cell cycle. Furthermore, the results of KEGG and Reactome analysis also confirmed that DEGs with the co-addition of H_2_O_2_ and NMN were mainly enriched in the cell cycle (Figure 5D,E; Appendix A).

Next, we found that 45 DEGs and 238 DEGs were specifically detected in group C vs. group H, and group C vs. group HN, respectively (Figure 6A). The 45 DEGs from group C vs. group H were displayed using the analysis of the heatmap (Figure 6B), and the top ten GO enrichment terms were also found to be related to molecular functions, including UDP-glycosyltransferase activity, acetylgalactosaminyl-O-glycosyl-glycoprotein beta-1,6-N-acetylglucosaminyltransferase activity, and pyrimidine deoxyribonucleoside binding (Figure 6C; Appendix A). Furthermore, the heatmap exhibited the DEGs expressed uniquely in group C vs. group HN (Figure 6D). Moreover, the top ten GO enrichment items were primarily related to the biological processes, such as response to lipids, single-organism process, and extracellular region part, most of which were involved in extracellular structure and stimulus responses that were critical to the cell cycle processes (Figure 6E; Appendix A). Next, the overlapped DEGs from group C vs. group H, and group C vs. group HN were analyzed (Figure 6F). Gene expression trends were also obtained, and those DEGs were mainly enriched in four kinds of trend (Figure 6G). More specifically, most DEGs were enriched in two kinds of trend, namely profile 1 and profile 6, which were then selected for KEGG analysis. To analyze the top fifteen KEGG pathways of profile 1 and profile 6, we found that the enriched pathways were the cell cycle, DNA replication, glutathione metabolism, and P53 signaling pathway, all of which were also mainly cell cycle-related pathways (Figure 6H,I; Appendix A).

### 2.5. NMN Partly Rescues the Cell Cycle Arrest, Apoptosis and Impaired Estrogen Synthesis of Ovarian Granulosa Induced by H_2_O_2_

To validate whether NMN can alleviate cell cycle arrest, apoptosis, and impaired estrogen synthesis in H_2_O_2_-cultured mGCs, the cells were supplemented with NMN simultaneously. Firstly, RT-qPCR was used for detecting the expression of selected DEGs. The results of RT-qPCR demonstrated that 150 μM H_2_O_2_ significantly increased the expression of genes related to oxidative stress, cell cycle, and apoptosis (e.g., *Homx1*, *Gsta4*, *Cdkn1a*, *P53*, and *Bax*), but decreased the expression level of antioxidative (e.g., *Bcl2*), proliferative (e.g., *Pcna* and *Cdk1*), and estrogen synthesis genes (e.g., *Cyp11a1*, *Cyp19a1*, *Star*, and *Hsd3a1*) (Figure 7A–C,F). Notably, 500 μM NMN could partially recover the expression levels of those genes, cell viability and E2 secretion of mGCs that were exposed to H_2_O_2_, and this indicate that NMN supplementation could partly repair H_2_O_2_-induced cell damages, such as cell cycle arrest and apoptosis, and estradiol secretion (Figure 7A–F). Furthermore, we validated protein levels of CYP19A1, BCL-2, BAX, and PCNA using Western blots, which were consistent with RT-qPCR results (Figure 7G).

Moreover, by using KGN as the other cell model, we then verified the above results from mGCs. When NMN (100, 300, or 500 μM) was supplemented to KGN that was exposed to H_2_O_2_ (100 or 200 μM), we observed that NMN supplementation could ameliorate H_2_O_2_-induced cell death (Figure 8A). Furthermore, the PI staining indicated that a high dose of NMN (300 and 500 μM) could partly alleviate the cell cycle arrest and apoptosis in KGN induced by 100 or 200 μM H_2_O_2_ (Figure 8B–E). Finally, considering that NMN exerts its biochemical role through converting into NAD^+^ in vivo to increase NADH levels, we detected the level of NAD^+^/NADH in KGN after the addition of H_2_O_2_ and/or NMN. The results showed that H_2_O_2_ (100 and 200 μM in Figure 8F and Figure 8G, respectively) significantly reduced the total amount of NAD^+^/NADH in KGN, whereas NMN (300 and 500 μM in Figure 8F and Figure 8G, respectively) supplementation significantly restored its levels (Figure 8F,G). This, therefore, suggests that NMN could partially alleviate the H_2_O_2_-induced cell cycle arrest and apoptosis by restoring the level of NAD ^+^/NADH in ovarian GCs.

## 3. Discussion

According to our analysis of ovarian GCs, the main findings of the present study are summarized as follows: (i) H_2_O_2_ exposure significantly changed the gene expression patterns in mGCs; (ii) H_2_O_2_ exposure negatively affected the cell cycle and oocyte maturation, inhibited proliferation, and induced apoptosis in ovarian GCs; (iii) NMN supplementation partly alleviated H_2_O_2_-induced cell cycle arrest and apoptosis in ovarian GCs. Therefore, we conclude that H_2_O_2_ significantly alters the gene expression pattern and cell cycle progression and induces apoptosis in ovarian GCs, but that NMN could partially alleviate these H_2_O_2_-induced abnormalities. These aspects will be discussed as follows.

Follicular development is the most vital process to determine the reproductive performance of female mammals [25]. The development of follicles originates in primordial follicles, which are oogonia derived from primordial germ cells through cell differentiation [26,27]. After the formation of primordial follicles, only a few primordial follicles can develop into mature follicles and then affect the homeostasis of reproduction [28]. The developmental process of primordial follicles can be divided into three stages: follicle recruitment, selection of dominant follicles, and development of dominant follicles before ovulation [29]. Non-dominant follicles eventually undergo atresia, which requires the involvement of apoptosis and autophagy [30]. However, abnormal apoptosis or autophagy in granulosa cells could result in excessive follicular atresia, and then lead to reproductive diseases, such as premature ovarian failure [31]. The current results show that H_2_O_2_ significantly altered the gene expression pattern in mGCs, inhibited proliferation, and induced apoptosis. These findings are consistent with previous results from human [32], pig [33], and bovine GCs [34]. Moreover, using RNA-seq, we revealed that the co-addition of H_2_O_2_ and NMN could increase the number of DEGs in mGCs, and this suggests that NMN supplementation positively shaped the gene expression pattern of H_2_O_2_-cultured mGCs.

The cell cycle is a complicated biological process, and the Cyclin is a protein family that regulates the cell cycle by activating cyclin-dependent kinase (CDK); both of them play a central role in the cell cycle regulation [35]. The progression of the cell cycle is necessary for cells to function properly, while cellular damage caused by oxidative stress can hinder the cell cycle and then lead to cell cycle arrest [14,15,16,17]. This is because the oxidative stress seriously affects the function of tissues or organs by inducing apoptosis [36,37]. ROS normally acts as a key factor in oxidative stress-induced apoptosis, which easily attacks biomolecules, and eventually causes apoptosis or necrosis [38,39]. Some studies have shown that these stimulations from physiological/pathological apoptosis further affect the progression of the cell cycle [40,41]. In this study, we observed that H_2_O_2_ exposure triggered the accumulation of β-galactosidase in mGCs. Meanwhile, the expressions of *P53* and *Bax* were also increased, but *Cdk1* was decreased significantly, thereby indicating that H_2_O_2_-oxidtaive stress could contribute to the onset of apoptosis. With respect to this, the underlying molecular mechanism might be that the DNA of cells was damaged by H_2_O_2_-induced oxidative stress. Generally, severe DNA damage directly breaks the cell cycle. When DNA damage is limited and reversible, the cells stop to proliferate, and some cells then enter the cell cycle upon exposure to DNA damage. However, the cells will rapidly undergo apoptosis when DNA damage is unrecoverable [42,43], and then stimulate expressions of P53, P16, and P21. Among them, p21 is a CDK inhibitor, which prevents the cell cycle to enter the S phase from the G1 phase by inhibiting the activity of the cyclin family. Of note, as a member of the INK4 family of CDK inhibitor (CKI), p16 protein not only inhibits the combination of the cyclin-dependent kinase CDK family and cyclin D, but also blocks the phosphorylation of pRb protein. Furthermore, this then causes the cell cycle to stop in the G1 phase, eventually leading to growth inhibition [44,45]. Indeed, we also found that adding NMN to the H_2_O_2_ condition significantly decreased the expression of *P53*, but significantly increased the expression of *Cdk1*. This, therefore, suggests that NMN partly reduces DNA damage caused by oxidative stress, hence alleviating cell cycle arrest and apoptosis.

Cellular oxidative damage negatively regulates aging-related signaling pathways and increases the expression of aging-related proteins, such as P53 and P21 [46,47]. The results from the KEGG analysis showed that one of the main enrichments of DEGs was the P53 pathway. Considering this, the expression of P53, a transcription factor, could be rapidly up-regulated upon the initiation of oxidative stress, and activate a downstream gene, such as p21, and then inhibit the phosphorylation of Rb protein, eventually causing cell apoptosis [14]. Furthermore, the results of CCK-8 and β-galactosidase staining also showed that H_2_O_2_ contributed to the aging of mGCs, and then reduced the cellular viability. Further screening of DEGs with the verification of RT-qPCR also showed that the expressions of genes, including *Cdkn1a*, *Gsta4*, and *Trp53inp1*, significantly increased in the presence of H_2_O_2_. Actually, overexpression of *Cdkn1a* (also known as P21) can lead to cell cycle arrest, and then results in a series of signs of cell aging [48]. Therefore, *Cdkn1a* is closely related to cellular aging, and it is recognized as the marker of cell aging.

Nicotinamide adenine dinucleotide (NAD^+^) is an important hydrogen carrier of oxidoreductase in cells [49], and it has direct effects on mitochondrial function and metabolism, immune response and inflammation, DNA repair and cell division [50,51]. NMN can only fulfill its physiological function by transforming into NAD^+^ in the body; this is because NAD^+^ plays an important role in maintaining the redox balance and regulating various metabolic activities of cells [52,53]. Recently, growing evidence has demonstrated that NMN could prevent un-controlled oxidative stress and increase the activity of aging cells [22,23]. Other research results demonstrate that NMN could alleviate programmed cell death, including apoptosis via NAD+/SIRT1 to reduce mitochondrial damage and endoplasmic reticulum stress [54,55]. In this study, ovarian granulosa cells encompassing mGCs and KGN upon H_2_O_2_ exposure led to declines in NAD^+^ levels, while NMN supplementation prevented it. Meanwhile, the results from RT-qPCR and Western blots also confirmed that NMN supplementation protected against H_2_O_2_-induced oxidative stress, which suggests that NMN could partially alleviate H_2_O_2_-induced cell cycle arrest, apoptosis, and impaired estrogen synthesis by restoring the levels of NAD^+^/NADH in ovarian GCs.

## 4. Materials and Methods

### 4.1. Mouse Granulosa Cell Collection

Three-week-old ICR mice (Chengdu Dashuo Laboratory Animal Co, Ltd., Chengdu, China) were intraperitoneally injected with 5 IU PMSG (Ningbo Second Hormone Factory, Ningbo, China), and the mice were sacrificed by cervical dislocation after 48 h. The mice were dissected, and their bilateral ovaries were removed in PBS supplemented with 1% penicillin/streptomycin (Invitrogen, Carlsbad, CA, USA). Next, the bursa and adipose tissues were removed under stereoscopic microscope. To collect granulosa cells (GCs), 1 mL syringes were used to puncture the follicles until the granulosa cells in the follicles were fully released. Suspension was then transferred to a 1.5 mL tube, the supernatant was discarded after centrifugation at 5000 r/min for 3 min. Then, PBS was added to resuspend the cells, and centrifuged again to remove the supernatant. Then, 0.25% trypsin (Beijing Solarbio Science & Technology Co., Ltd., Beijing, China) was added to digest cells for 4 min, and cell culture medium was then added to terminate the digestion. After centrifugation, the supernatant was discarded and the cells were resuspended in the cell culture medium, cultivated to 6 cm petri dishes, and incubated in a 37 °C, 5% CO_2_ incubator. The cell culture medium was supplemented with 10% fetal bovine serum (Newzerum, Christchurch, New Zealand) and 1% penicillin/streptomycin based on DMEM-F12 medium (Ranjeck Technology Co., Ltd., Beijing, China).

### 4.2. GCs Cell Culture

GCs were placed in the CO_2_ incubator for 6–12 h; then, the medium in the petri dish was discarded. Next, the cells were rinsed with PBS supplemented with 1% penicillin/streptomycin and the PBS was discarded. Then, 1 mL of 0.25% trypsin was added to the dish to digest the cells for 3 min in incubator. After that, the morphology of cells was determined under a stereomicroscope. Then, 0.5 mL of cell culture medium was added to terminate the digestion. Next, the cells were collected into a 1.5 mL tube and centrifuged at 5000 r/min for 3 min. After the removal of the supernatant, the cells were resuspended in 1 mL of fresh cell culture medium and cultivated in new petri dishes or wells. For the human granulosa cell line (KGN), the cells were cultured in the above cell culture medium and passaged every 2 days. When the cell confluence reached 70%, the basal medium containing drugs was replaced to treat the cells. When drug treatments were completed, the cells were collected by Trizol reagent (Invitrogen, Carlsbad, CA, USA) for RNA extraction, or fixed by 4% paraformaldehyde for staining.

### 4.3. High-Throughput RNA Sequencing and Bioinformatic Analysis

In the present study, we designed three different groups: group C (control), group H (150 μM H_2_O_2_), and group HN (150 μM H_2_O_2_ + 500 μM NMN). The samples were sent to Gene Denovo Biotechnology Co., Ltd. (Guangzhou, China) for RNA-Seq sequencing. Briefly, RNA samples were processed by the following methods. Firstly, total RNA was extracted using Trizol reagent kit. RNA quality was assessed on a bioanalyzer and detected using RNase free agarose gel-electrophoresis. Then, eukaryotic mRNA was enriched by Oligo (dT) beads. It was then transcribed into cDNA in the opposite direction, and the full-length cDNA was repaired at the end to construct a library. The resulting cDNA library was sequenced using Illumina Novaseq6000. In order to obtain high-quality clean reads, the resulting data were filtered, and the low-quality data containing connectors were removed to obtain the high-quality data.

To screen the DEGs, the genes/transcripts with the parameters of false discovery rate (FDR) below 0.05 and absolute fold change ≥ 2 (2-fold) were used to filter DEGs for the biological analysis. The Database for Annotation, Visualization, and Integrated Discovery (DAVID v6.7; http://david.abcc.ncifcrf.gov, accessed on 8 January 2023) was used for annotating biological themes (Gene ontology, GO). The Kyoto Encyclopedia of Genes and Genomes (KEGG; http://www.genome.jp/kegg/, accessed on 8 January 2023) was used for determining the associated pathways. The Reactome (Reactome; https://reactome.org/, accessed on 8 January 2023) was used to influx species reactions and biological pathways.

All transcriptome data obtained in this study have been uploaded to NCBI (BioProject: PRJNA930590).

### 4.4. Real-Time Quantitative Polymerase Chain Reaction (RT-qPCR)

The RT-qPCR was conducted according to our previous study [56]. Briefly, specific genes were screened for RT-qPCR verification. The reaction system was 10 μL, including 5 μL SYBR prime Ex Taq TM II, 1 μL cDNA template, 0.5 μL 10 μM PCR Forward primer and Reverse primer, and 3.0 μL RNase free dH_2_O. The reaction conditions were as follows: pre-denaturation at 94 °C for 30 s; denaturation at 94 °C for 10 s and annealing at 60 °C for 15 s, 39 cycles of denaturation and annealing were repeated; 65 °C for 5 s, the melting curve of 65~95 °C increased by 5 s at 0.5 °C. Each gene was repeated three times, and *GAPDH* was used as the internal reference gene. All primers used in the present study were presented in Table 1.

### 4.5. Cell SA-β-Gal Staining

Senescence β-Galactosidase Staining Kit (Beyotime Biotechnology, Shanghai, China) was used for detecting GCs status. First, after the cells were cultured for 24 h and the culture medium was completely removed. The cells were washed once with PBS and 1 mL β-galactosidase staining fixative was then added. After fixation at room temperature for 15 min, the cell fixative was removed, and the cells were washed with PBS three times. PBS was removed and 1 mL staining solution was then added to each well. The cells were incubated overnight at 37 °C, and the staining pictures were captured under a microscope.

### 4.6. Western Blots

In this experiment, 150 μM H_2_O_2_ and/or 500 μM NMN were used to treat mGCs in group H and group HN. After 24 h treatments, mGCs were collected to extract total protein. Using the BSA kit, we determined protein concentrations in different groups. Then, a total of 15 μg protein in each sample were used to detect target proteins. Proteins were transferred into PVDF membrane from 12% SDS-PAGE after 1 h electrophoresis under 120 V constant voltage, and blocked in 1% BSA solution for 1 h. Diluent primary antibodies (CYP19A1 (Bioworld, Louis Park, MN, USA); BCL-2 (Affinity Biosciences, Changzhou, China); BAX (Affinity Biosciences, Changzhou, China); PCNA (Affinity Biosciences, Changzhou, China); β-ACTIN (Cell Signaling Technology, Danvers, MA, USA)) were then added to membrane and incubated in 4 °C overnight. Membranes were rinsed in TBST three times. Then, membranes were immersed in diluent second antibodies (IgG(H + L) (Beyotime Biotechnology, Shanghai, China) at room temperature for 1 h. Again, membranes were rinsed in TBST three times. An ECL kit (Beyotime Biotechnology, Shanghai, China) was used for visualizing protein bonds, and a chemical exposure instrument was used for capture pictures.

### 4.7. The PI Staining

The Calcein/PI Cell Viability/Cytotoxicity Assay Kit (Beyotime Biotechnology, Shanghai, China) was used for live cell counting. According to the instructions, the cells were cultured for 24 h and inoculated on a 96-well plate. The cell culture medium was taken away, and then washed by PBS once. Each well was added with 100 μL Calcein AM/PI detection working solution, and then incubated at 37 °C in the dark for 30 min. After the incubation, the staining result was observed under a fluorescence microscope. The whole process was performed in the darkness.

### 4.8. Total NAD^+^/NADH Detection

The NAD^+^/NADH detection kit (Beyotime Biotechnology, Shanghai, China) was used for determining the total amount of NAD^+^/NADH in cells. The medium in cell dish was discarded and then rinsed by PBS once. After that, the culture medium was removed, and 200 μL ice-cold NAD^+^/NADH extraction buffer was then added. Cells were pipetted gently to promote cell lysis on ice. Tubes were then centrifuged at 12,000× *g* for 10 min at 4 °C. The supernatant was collected and kept on ice for subsequent assay. Briefly, 60 μL supernatant was pipetted into a centrifuge tube, and then incubated for 30 min at 60 °C in a PCR thermocycler to degrade NAD^+^. The mixture in each well was vortexed after adding alcohol dehydrogenase working solution, and the formation of air bubbles was avoided. Samples were incubated at 37 °C for 10 min in the dark and 10 μL developer solution was then added into each well, and the samples were incubated at 37 °C for 30 min in the dark. The absorbance at 450 nm was measured. The average absorbance of each point with the standard group was calculated. The standard curve was drawn with the concentration of NADH as the abscissa and the absorbance as the ordinate.

### 4.9. Cell Viability Determination

The CCK-8 Cell Proliferation and Cytotoxicity Assay Kit (Biosharp, Hefei, China) was used for detecting the cell viability. Briefly, the cells in the medium were seeded at a density of 8 × 10^4^ cells/well in 96-well plates. After 24 h, the cells were sampled according to the above grouping, respectively. Then, CCK-8 solution was added into each well. Followed by 1 h incubation, the optical density (OD) at 450 nm was determined. Cell viability = (A experimental group − A blank hole)/(A normal control group − A blank hole) × 100%.

### 4.10. Detection of E2 Level

Mouse E2 detection kit (Jianglaibio, Shanghai, China) was used for determining the E2 level in each group according to the manufacturer’s instructions.

### 4.11. Statistical Analysis

All experiments were replicated at least three times. Results were represented as means ± standard deviation (SD), and then analyzed with Student’s *t*-test or one-way analysis of variance using the GraphPad Prism software 8.0.2 (GraphPad Software Inc., San Diego, CA, USA). The *p*-value below 0.05 was considered as the threshold of significant difference.

## 5. Conclusions

On the basis of the present findings from ovarian GCs, we indicate that NMN supplementation remarkably changes the gene expression pattern and cell cycle progression of H_2_O_2_-cultured mGCs. More specifically, H_2_O_2_ exposure negatively affects the gene expression pattern of ovarian GCs by arresting the cell cycle progression, inhibiting proliferation, and triggering apoptosis, while NMN supplementation partly alleviate these H_2_O_2_-induced abnormalities.

## Figures and Tables

**Figure 1 ijms-24-15666-f001:**
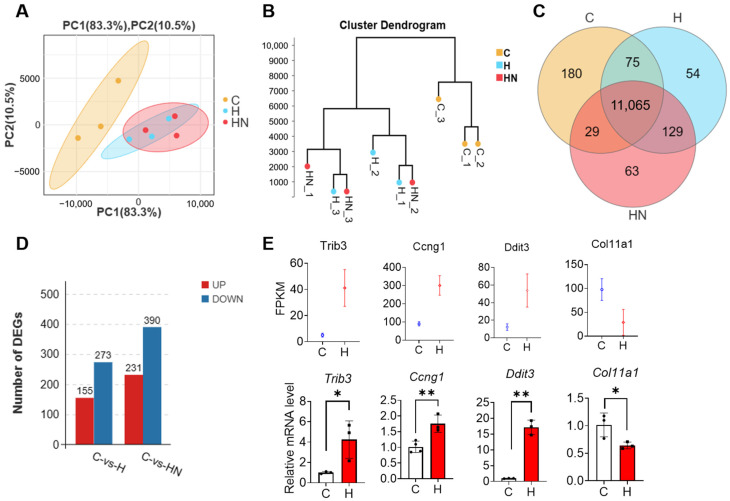
The effects of NMN supplementation on gene expression pattern in mouse granulosa cells (mGCs) that are exposed to hydrogen peroxide (H_2_O_2_). (**A**) The principal component analysis (PCA) of group C (control), group H (150 μM H_2_O_2_), and group HN (150 μM H_2_O_2_ and 500 μM NMN). (**B**) The cluster analysis of group C, group H, and group HN. (**C**) Venn diagram of gene abundance detected in the three groups above. (**D**) Number of differentially expressed genes (DEGs). (**E**) The verification for DEGs by the RT-qPCR method. Each value represents mean ± SD. * *p* < 0.05, ** *p* < 0.01.

**Figure 2 ijms-24-15666-f002:**
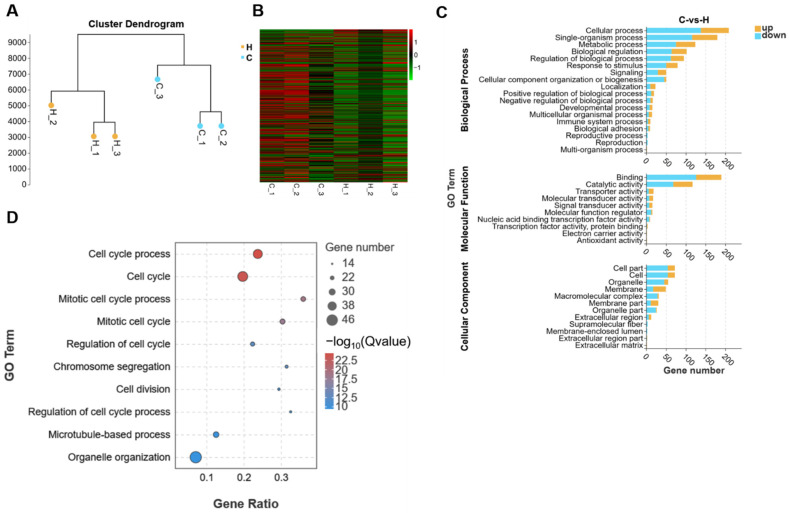
RNA-seq analysis revealed that H_2_O_2_ induced mGC cell cycle arrest and apoptosis. (**A**) The cluster analysis of group C (control) and group H (150 μM H_2_O_2_). (**B**) The heatmap of DEGs between the two groups. (**C**) The GO classification of DEGs. (**D**) The GO enrichment analysis of DEGs.

**Figure 3 ijms-24-15666-f003:**
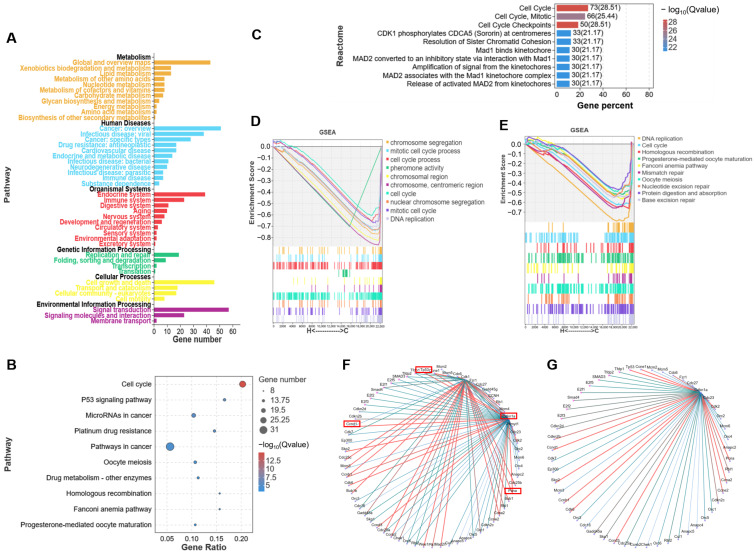
RNA-seq sequencing analysis revealed that H_2_O_2_ triggered cell cycle arrest and apoptosis in mGCs. (**A**) The KEGG classification of DEGs between group C (control) and group H (150 μM H_2_O_2_). (**B**) The KEGG enrichment analysis of DEGs between the two groups. (**C**) The Reactome database analysis of DEGs. (**D**) The GO-GSEA analysis of genes. (**E**) The KEGG-GSEA analysis of genes. (**F**) The protein interaction network of *Cdk1* gene. (**G**) The protein interaction network of *Cdkn1a* gene.

**Figure 4 ijms-24-15666-f004:**
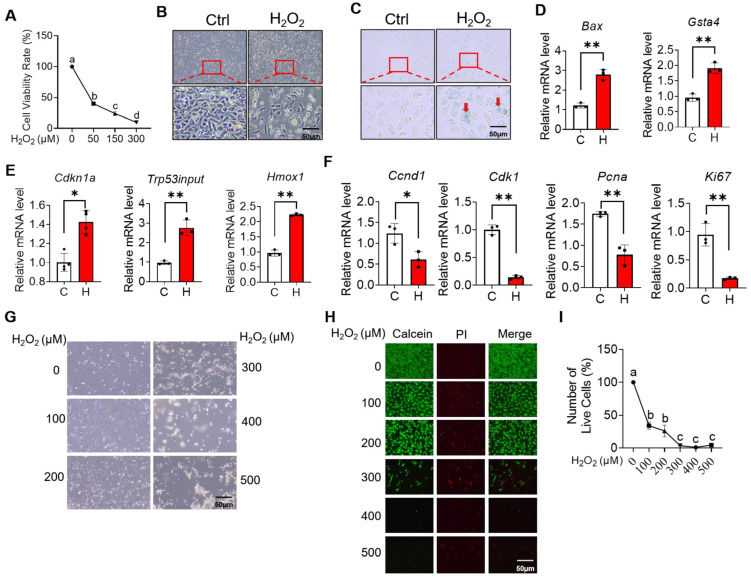
H_2_O_2_-induced ovarian GCs cell cycle arrest and apoptosis. (**A**) The cell viability rate of mGCs detected by CCK8. (**B**) Bright field images of mGCs suffering 200 μM H_2_O_2_. (**C**) β-galactosidase staining pictures of mGCs treated with 200 μM H_2_O_2_. Red arrows represent β-galactosidase positive cells. (**D**–**F**) mRNA expression of genes related to antioxidant (**D**), apoptosis and ferroptosis (**E**), and proliferation (**F**). The mGCs in group H were treated with 150 μM H_2_O_2_. (**G**) The morphology of human granulosa cells (KGN) that were treated with different concentrations of H_2_O_2_ under bright field. (**H**) The PI staining of KGN cells that were cultured with different concentrations of H_2_O_2_. (**I**) Number of living cell of KGN cells that were treated with different concentrations of H_2_O_2_. Data represent mean ± SD. * *p* < 0.05, ** *p* < 0.01. Different letters upon panels mean significant difference.

**Figure 5 ijms-24-15666-f005:**
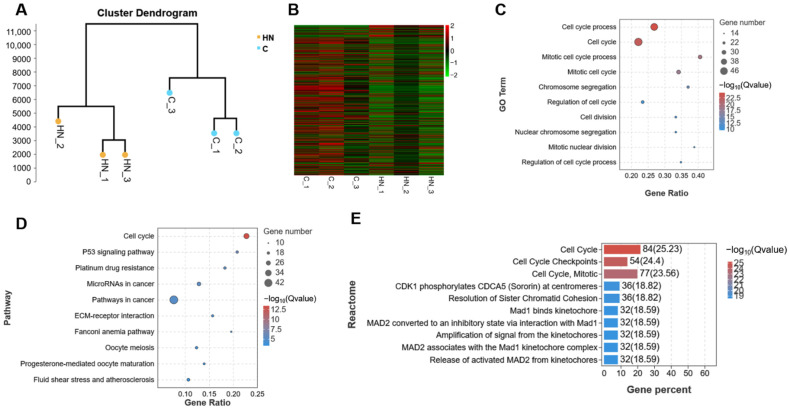
RNA-seq revealed that co-addition of H_2_O_2_ and NMN changed the gene expression pattern in mGCs. (**A**) The cluster analysis of group C (control) and group HN (150 μM H_2_O_2_ + 500 μM NMN). (**B**) The heatmap of DEGs comparing group C and group HN. (**C**) The GO enrichment of DEGs. (**D**) The KEGG enrichment of DEGs. (**E**) The Reactome database analysis of DEGs.

**Figure 6 ijms-24-15666-f006:**
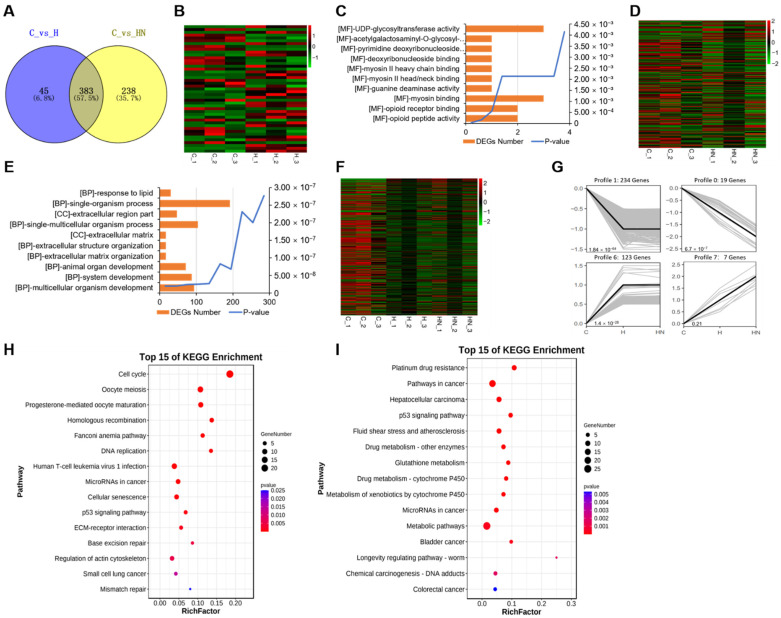
RNA-seq sequencing revealed that the co-addition of H_2_O_2_ and NMN changed gene expression patterns in mGCs. (**A**) The Venn diagram of DEGs comparing group C (control) and the group H (150 μM H_2_O_2_), and group C and the group HN (150 μM H_2_O_2_ + 500 μM NMN). (**B**) The heatmap of unique DEGs in group C vs. group H. (**C**) The top 10 GO items enriched by unique DEGs in group C vs. group H. (**D**) The heatmap of unique DEGs in group C vs. group HN. (**E**) The top 10 GO items enriched by unique DEGs in group C vs. group HN. (**F**) The heatmap of overlapped DEGs between group C and group H, and group C and group HN. (**G**) The trend analysis of overlapped DEGs between group C and group H, and group C and group HN. (**H**) The top 15 KEGG pathways of DEGs in profile 1. (**I**) The top 15 KEGG pathways of DEGs in profile 6.

**Figure 7 ijms-24-15666-f007:**
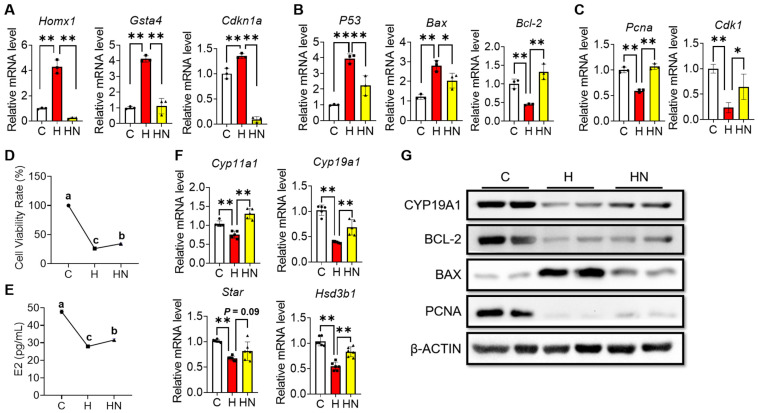
NMN partly alleviated H_2_O_2_-induced cell cycle arrest, apoptosis, and impaired estrogen synthesis in mGCs. (**A**–**C**) The mRNA expressions of antioxidative (**A**), apoptosis (**B**), and cell cycle genes (**C**). mGCs in group H that were treated with 150 μM H_2_O_2_ for 24 h. The mGCs in group HN that were treated with 150 μM H_2_O_2_ and 500 μM NMN for 24 h. (**D**) Cell viability of mGCs in group C, group H, and group HN. (**E**) Estradiol (E2) levels of mGCs in three groups detected by ELISA kit. (**F**) The mRNA expressions of estrogen synthesis genes in three groups. (**G**) Protein levels of CYP19A1, BCL-2, BAX, and PCNA in three groups detected by Western blots. Each value represents mean ± SD. * *p* < 0.05, ** *p* < 0.01. Different letters upon panels mean significant difference.

**Figure 8 ijms-24-15666-f008:**
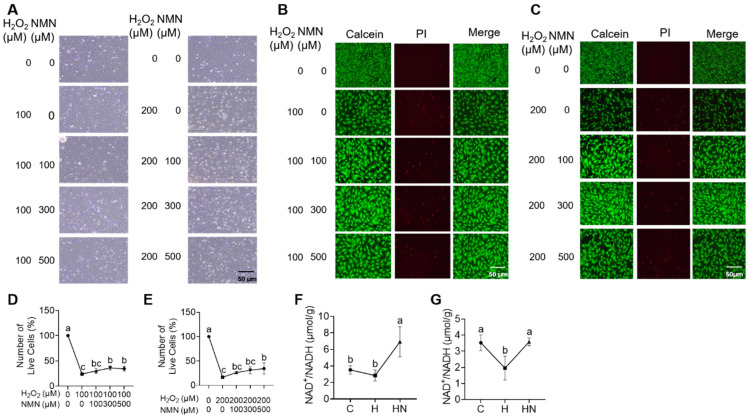
NMN partly alleviated H_2_O_2_-induced cell cycle arrest and apoptosis in KGN. (**A**) Bright field results of KGN treated by H_2_O_2_ (100 or 200 μM) and/or NMN (100, 300, or 500 μM). (**B**,**D**) The PI staining (**B**) and statistical results (**D**) of KGN cells treated with different concentrations of NMN in the presence of 100 μM H_2_O_2_. (**C**,**E**) The PI staining (**C**) and statistical results (**E**) of KGN cells treated with different concentrations of NMN in the presence of 200 μM H_2_O_2_. (**F**,**G**) Detection of the total amount of NAD^+^/NADH in group C, group H, or group HN in the presence of 100 μM (**F**) or 200 μM (**G**) H_2_O_2_ and 300 μM (**F**) or 500 μM (**G**) NMN. Each value represents mean ± SD. Different letters upon panels mean significant difference.

**Table 1 ijms-24-15666-t001:** Primers for RT-qPCR used in this study.

Gene	Primers	Sequences (5′-3′)
*Cdkn1a*	Forward	TCCAGACATTCAGAGCCACAG
	Reversed	AAAGTTCCACCGTTCTCGGG
*Hmox1*	Forward	GCCCCACCAAGTTCAAACAG
	Reversed	GCTCCTCAAACAGCTCAATGT
*Gsta4*	Forward	ACTTTAATGGCAGGGGACGG
	Reversed	GCAGGTGTCCATCCTTTTGC
*Ccng1*	Forward	AAGACGTGGCTGTCAAGATGA
	Reversed	TGTCTCCGTGTCAAAGCCAA
*Trp53inp1*	Forward	TAAGACTCACGGGCACAGAAATG
	Reversed	AGTGTGGCAATCCCTGGTAAG
*Gapdh*	Forward	TGCCCCCATGTTTGTGATG
	Reversed	TGTGGTCATGAGCCCTTCC
*Ddit3*	Forward	CCTGAGGAGAGAGTGTTCCAG
	Reversed	GACACCGTCTCCAAGGTGAA
*Col11a1*	Forward	ACAGTAGCACAAACAGAGGCAA
	Reversed	AATCCCTGCCGTCTACTCCT
*P53*	Forward	TGATGGAGAGTATTTCACCC
	Reversed	GGGCATCCTTTAACTCTAAGA
*Bax*	Forward	CGGCGAATTGGAGATGAACTG
	Reversed	GCAAAGTAGAAGAGGGCAACC
*Cdk1*	Forward	CGGTACTTACGGTGTGGTGTAT
	Reversed	CTCGCTTTCAAGTCTGATCTTCT
*Ccnd1*	Forward	AGGGATGATGATGCTGGTATG
	Reversed	AACACCACACCTGGGCTTAT
*Pcna*	Forward	TAAAGAAGAGGAGGCGGTAA
	Reversed	TAAGTGTCCCATGTCAGCAA
*Ki67*	Forward	AATCCAACTCAAGTAAACGGGG
	Reversed	TTGGCTTGCTTCCATCCTCA
*β-action*	Forward	GCAGAAGGAGATCACTGCCCT
	Reversed	GCTGATCCACATCTGCTGGAA
*Cyp11a1*	Forward	ACCAAGAACTTTTTGCCCCT
	Reversed	ATGTCCCCCGAGTAATTTCC
*Cyp19a1*	Forward	GACTTTGCCACTGAGTTGATTT
	Reversed	CGATCAGCATTTCCAATATGCA
*Star*	Forward	ACGTGGATTAACCAGGTTCG
	Reversed	CAGCCCTCTTGGTTGCTAAG
*Hsd3b1*	Forward	CTCTTCTGTCCAGCTTTTAAC
	Forward	ACCAAGAACTTTTTGCCCCT

## Data Availability

Data used or analyzed during the study are available from the corresponding author upon reasonable request.

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
