# Peer review of "β-Nicotinamide Mononucleotide Alleviates Hydrogen Peroxide-Induced Cell Cycle Arrest and Death in Ovarian Granulosa Cells"

_ijms, 2023, doi:10.3390/ijms242115666_

Round 1
Reviewer 1 Report
Β-nicotinamide Mononucleotide Alleviates Hydrogen Peroxide-induced Cell Cycle Arrest and Death in Ovarian Granulosa Cells
In this study, Wang et al. determined the effects of NMN on gene expression patterns and antioxidant capacity using two types of ovarian GCs — mouse GCs (mGCs) and human granulosa cell line (KGN) — exposed to hydrogen peroxide. They concluded that “H2O2 and/or NMN significantly changed the gene expression patterns in mGCs; H2O2 negatively affected cell cycle and oocyte maturation, inhibited proliferation, and induced apoptosis in ovarian GCs; and NMN supplementation partly rescued H2O2-induced cell cycle arrest and apoptosis in ovarian GCs.” This is an interesting study; the results are presented clearly, and the manuscript is mostly well-structured and well-written. This reviewer feels this manuscript has potential for improvement and recommends a revision. Please consider the following comments/suggestions for the revision.
1. In Figures 1, 2, 3, 4 (panels B-F), 5, 6, and 7 (panels A-D), the concentrations of H2O2 and/or NMN are unclear. Please mention the concentrations of the drugs used to treat the samples used for the transcriptome analysis (both in the figure legends and the results/methods sections). It is possible that different gene expression patterns are expressed at lower H2O2 concentrations compared to higher H2O2 concentrations. How would the authors address this, and what do the authors speculate about the differences in the effect of NMN on cells treated with lower H2O2 doses?
2. The authors have shown the changes in gene expression using RNA-seq. It would be nice if the changes at the protein level were also shown (either by western blot or immunocytometry).
3. In Figure 7, the order of treatment is unclear. Were the cells pretreated with NMN and then exposed to H2O2, or was a different treatment protocol followed?
4. The PI panels in Figures 3H, 7F, and 7G are not very clear. Please consider providing better images.
5. Also, the differences in the cell numbers between the H and HN groups are not very clear in the micrographs (from the Calcein/PI staining images) in Figures 7F and 7G, although the quantification plots show a significant effect. Although the quantification showed a significant difference, the biological significance of the effect of NMN is uncertain. It would be interesting to see the effect of NMN on cells treated with lower doses of H2O2.
6. The authors have used the CCK method to determine cell viability. This method is based on the activity of mitochondrial enzymes, and using it to determine cell viability after inducing significant oxidative stress (high H2O2 doses) could result in non-accurate results. I recommend that the authors use other methods for determining cell viability. A flow cytometric determination of cell viability/cell cycle could be more informative.
The English language in the manuscript could be improved. There are several minor errors/typos in the manuscript. I have pointed out a few below.
1. Line 67 – Wrong symbol for comma (,)
2. Line 69 – ‘have been proved’ to ‘have been proven’
3. Line 96 – ‘And, we then analyzed all groups by the cluster analysis, it also displayed that the control group was significantly distinct.’ ‘Cluster analysis also showed that the control group significantly differed from the treatment groups.’
4. Line 121 – unnecessary use of the article ‘the’
5. Line 127 – DEGs misspelled as DGEs
6. Line 161 – ‘all of enriched’ to ‘all enriched’
7. Line 184 – ‘with H2O2 concentration-dependent manner” to ‘in an H2O2 concentration-dependent manner.’
8. Line 198 – ‘microscopy’ instead of ‘photography’
9. Line 201 – ‘but again with the number of floating dead cells’ – I'm not sure what the authors were trying to say here.
10. Line 222 – ‘investigated’ instead of ‘investigate’
Reviewer 2 Report
In the study entitled "β-nicotinamide mononucleotide alleviates hydrogen peroxide induced cell cycle arrest and death in ovarian granulosa cells", the authors have conducted a series of experiment to evaluate effect of H2O2 on molecular and cellular changes in mouse granulosa cells. They further show that supplementation with NMN can alleviate some of the deleterious effects of H2O2 induced oxidation stress.
The manuscript that i received didnt have supplemental data attached so I couldnt compare changes in the expression of key genes between control and H group vs C and HN group. Besides that major concerns that need to be addressed include:
1) Authors did not mention what was the concentration of H2O2 and NMN used for transcriptomic studies.
2) Line 89-91: Principal component of what? Is it total variance between C and H?
3) Figure 1A " The PCA analysis clusters H and HN group together, please reflect on this observation.
4) Line 96: which genotypes are the authors talking about?
5) Lines 168-170: How was the corelation between hub genes and cell cycle genes calculated and confirmed?
6) For validation of RNA-seq results, why were random genes chosen for PCR analysis instead of genes associated with, e.g, oxidation stress response.
7) Lines 250-252: How are BPs listed here involved in cell cycle progression.
8) Discussion: Discussion is vague, needs more elaboration on meaning of results and discussion of a link between NMN and alleviation of apoptosis.
Authors mentioned hormonal dysregulation as a consequence of oxidation stress, it will be benifical to evaluate if H2O2 causes changes in estradiol secretion in culture medium and whether that can be reversed by the supplementation with NMN.
Minor comments:
Line 121: group C and the group H
Line 125: DEGs are shown in
Line 142: showed that those DEGs were.... Which DEGs are the authors talking about.
Lines 147-148 and Line 360: Please give Reference.
Figure 2: Please define colors used for visualization.
Line 346: When DNA damage is limited/less
I am not an expert in English language, but though the manuscript is understandable, moderate editing and rephrasing (in some parts) is required, especially in Materials and Methods, Results and Discussion sections. Some examples (errors are not limied to these) are:
Line 16: estrogen secretion by follicle...
Line 43: flattened somatic cells surrounding oocytes multiply to form several layers accompanied with the increase in the cell numbers.
Lines 47-49: needs Reference
Line 74: functional decline..
Line 184-185: decrease with increasing H2O2 concentration.
e.g., Lines 89-91, p5-96, 106-107 need rephrasing. In lines 200-201, it is not clear what authors want to convey.
Round 2
Reviewer 2 Report
I am satisfied with changes that authors have made.
Some minor comments.
How did authors select concentration of NMN
Line 44: please remove are
Line 95: please remove and
Please rephrase lines 364-366
Manuscript will benefit from a careful reading and correction of minor mistakes